# Effect of *Spirulina platensis* Supplementation on Carcass Characteristics, Fatty Acid Profile, and Meat Quality of Omani Goats

**DOI:** 10.3390/ani13182976

**Published:** 2023-09-20

**Authors:** Fahad Al-Yahyaey, Waleed Al-Marzooqi, Ihab Shaat, Melanie A. Smith, Jamal Al-Sabahi, Sherif Melak, Russell D. Bush

**Affiliations:** 1Animal Nutrition Research Division, Ministry of Agriculture Wealth, Fisheries and Water Resources, Rumais, P.O. Box 467, Muscat 100, Oman; 2School of Veterinary Science, Faculty of Science, University of Sydney, Camden, NSW 2570, Australia; msmith@mla.com.au (M.A.S.); russell.bush@sydney.edu.au (R.D.B.); 3Animal and Veterinary Sciences Department, College of Agricultural & Marine Sciences, Sultan Qaboos University, Al-Khod, P.O. Box 34, Muscat 123, Oman; walmar@squ.edu.om; 4Oman Animal and Plant Genetic Resources Centre, Ministry of Higher Education, Research and Innovation, Al Koudh, P.O. Box 92, Muscat 123, Oman; shaat@hotmail.com; 5Animal Production Research Institute, Agriculture Research Center, Ministry of Agriculture, Dokki, Giza 12618, Egypt; melakshaa@yahoo.com; 6Central Instrument Laboratory, College of Agricultural & Marine Sciences, Sultan Qaboos University, Al-Khod, P.O. Box 34, Muscat 123, Oman; jsabahi98@gmail.com

**Keywords:** carcass traits, meat quality, fatty acids, spirulina, Jabbali and Sahrawi goats

## Abstract

**Simple Summary:**

Current low meat production in Omani goats under traditional feeding regimes justifies a study on the effects of Spirulina platensis (SP) supplementation on goats’ performance. Hence, this study aimed to evaluate the effect of incorporating SP into diets on the carcass characteristics, fatty acid profiles, and meat quality traits of two Omani goat breeds. The dietary inclusion of SP at daily doses of 2 g and 4 g/goat can enhance weight gain, meat quality, and fatty acid profiles without affecting their physiological health. However, further research may be needed to evaluate the effects of different doses of SP supplementation on the diet.

**Abstract:**

In a 70-day study, 36 Jabbali and Sahrawi bucks, aged 11 months, were utilized to evaluate the effects of different levels of spirulina dietary supplement (SP) on carcass characteristics, fatty acid profile, and meat quality traits in Omani goat breeds. The goats were put into six groups of six bucks, each at random. The diet consisted of a conventional concentrate feed ration (CFR) without spirulina (CON), and the CFR diet supplemented with spirulina at the levels of 2 g/head daily (T1) and 4 g/head daily (T2). In general, Sahrawi bucks showed a highly significant response to SP feeding compared with Jabbali bucks. The treatment groups, especially T1, showed a significant increase in average daily gain and carcass traits (body length, leg length, and the rack weight) compared with the CON group of Sahrawi bucks. The weights of omental and kidney fat were also significantly higher in T1 compared with CON and T2 groups of Sahrawi goats, while they were significantly higher in T2 compared with CON and T1 groups of Jabbali goats. Carcass profile and meat quality, including ultimate ph and meat color lightness (L*) were increased significantly with dietary spirulina in both LD and SM muscles of Sahrawi goats. Most of the Sfa, Mufa, Pufa, Pufa n-6, Pufa n-3, and n-6/n-3 ratios of the LD showed significant differences in diets supplemented with SP compared with CON for Sahrawi bucks, while some of them were significant in Jabbali bucks. The LD muscle of Sahrawi goats fed diets supplemented with SP of the T1 group significantly decreased in the amounts of pentadecanoic and margaric acids compared with the T2 and CON groups. The study concluded that incorporating SP (2 g and 4 g/head daily) into the diet of Omani goats, especially Sahrawi goats, can increase growth performance, as well as improve fatty acid composition and meat quality.

## 1. Introduction

Goat meat is commonly produced in many parts of the world, especially in Asia (571 million heads) and Africa (481 million heads), which accounts for 95% of the total world’s goat meat production (1.1 billion). Goat meat production increased globally by 25% from 2011 to 2021 [1], “However, the global goat meat industry is challenged by rising input costs, making it imperative to continuously enhance goat production and improve meat quality standards to remain competitive in the global market and meet the evolving demands of the health-conscious consumer” [2]. The impact of nutrition on meat quality and nutritional content is widely acknowledged and is well documented. Changes in the diet and breeding strategies can significantly alter biological characteristics of muscle tissue, affecting its quality and nutritional value [3]. This relationship between nutrition and meat quality was demonstrated in several studies [4]. Adeyemi et al. [5] stated that one of the effective strategies to improve the fatty acid profile of meat is by increasing the antioxidant profile of ruminant diets using a dietary mix of 20% palm oil and 80% canola oil to align with the evolving demands and preferences of consumers. After the COVID-19 pandemic, the search for low-cost, long-term alternatives to the requirements for animal feed emerged as a critical research topic of highest priority and strategic aim [6].

Over the past decade, there was extensive advertising worldwide about the beneficial effects of feeding microalgae, leading to the proliferation of algae production enterprises at both small and large levels of production [7,8]. Supplementing the diets of small ruminant farming with alternative antioxidant sources, such as *Spirulina platensis* (SP), holds a great promise in elevating the standard of goat meat and its nutritional composition.

SP is a blue-green algae that offers a range of health-beneficial chemicals due to its rich chemical composition. It is widely used as a dietary supplement for livestock as it contains a high percentage of protein (60–70%) and total lipids content (5–6%) along with substantial amounts of vitamins, minerals, chlorophyll, carotenoids, carbohydrates, sterols, and pigments such as phycocyanin and allophycocyanin [9,10,11,12]. SP proved to be a remarkable discovery with significant biological and economic implications. It has a broad range of applications across various industries, including food, pharmaceuticals, biofuels, cosmetics, and agriculture [9]. Habib [13] and AlFadhly et al. [9] showed that SP is rich in polyunsaturated fatty acids (Pufa) and other health-promoting fatty acids, including gamma-linoleic acid, eicosapentaenoic acid (EPA) and docosahexaenoic acid (DHA). Furthermore, SP was found to possess numerous health benefits, including antiviral, antioxidant, hepatoprotective, anti-allergenic, carotenoids and immunomodulatory properties [14,15,16]. In addition, SP is commonly recommended as a dietary protein supplement that can improve growth rate and meat quality in ruminants, chickens, pigs, and rabbits [17,18,19].

Meat quality is an important factor influencing consumer eating quality parameters and acceptance of the product [2]. Whilst goat meat is widely consumed globally, its consumption is smaller compared to other red meats such as beef and sheep meat [20]. Pophiwa et al. [2] reported that poor goat meat quality is mostly caused by the use of inappropriate animals, poor nutrition, and inadequate pre- and post-slaughter handling practices. In Oman, goat meat is the preferred red meat of choice [21], with consumers eating it 14 times a week on average [20]. However, poor nutrition and lack of managed breeding are the primary reasons for low levels of production in goats under traditional systems in Oman, where animals are grown and finished mostly on low-quality rangeland pastures [22]. The production of Omani goat meat and meat quality requires more research to determine the response to SP supplementation to fill this gap. Therefore, this study aims to evaluate the effect of feeding SP diets on the carcass characteristics, fatty acid profile, and meat quality of Jabbali and Sahrawi Omani goat breeds that were reared under intensive farming.

## 2. Materials and Methods

### 2.1. Ethics Statement

The study was performed at the Livestock Research Center, Directorate General of Agriculture and Livestock Research, Ministry of Agriculture, Fisheries and Water Resources, Muscat, Sultanate of Oman. Ethics were approved by the University of Sydney Animal Ethics Committee (AEC) 2019/1597 according to the New South Wales (NSW) Animal Research Act 1985 and its associated regulations, the Australian Code for the care and use of animals for scientific purposes 8th Edition 2013, and the Australian Code for the Responsible Conduct of Research 2007.

### 2.2. Animals Management and Diets

Thirty-six eleven-month-old bucks (average body weight: 16.44 ± 0.33 kg) of two main domestic breeds of goats in Oman, Jabbali (*n* = 18) and Sahrawi (*n* = 18), were randomized according to a true experimental design to one of three feeding treatments including Control (CON), Treatment 1 (T1), and Treatment 2 (T2) in a 2 × 3 factorial arrangement (*n* = 6 per group) for 70 days. The CON group was fed a conventional concentrate ration (14% crude protein and 11.97% energy MJ/kg DM). Animals in T1 and T2 were fed the control ration with an addition of 2 g and 4 g/head/day of SP, respectively. The basal diet was formulated to meet the goats’ nutrient requirements to achieve a body weight gain at a rate of 0.3 kg/day. All animals were fed twice a day at 8:00 a.m. and 3:00 p.m. Goats were fed commercial SP pellets (DXN International Australia Pty. Ltd., Parramatta, NSW, Australia) daily at 8:00 am. All animals were individually housed in randomly allocated pens throughout the experiment, including two weeks of acclimatization, during the winter season (November–February 2019–2020). To provide roughage together with free access to freshwater during the experimental period, Rhodes grass (*Chloris guyana*) was fed ad libitum. Feed intake was calculated as the difference between the feed offered and feed residual amounts, and the feed residual was obtained and recorded in the morning of the next day. Table 1 presents the nutritional and dry matter composition of ingredients in the experimental diets. Appendix A presents the basal diet components mean fatty acid composition (% total FA). The proximate compositions of the experimental diet were determined using the methods described by the Association of Official Analytical Chemists (AOAC) [23].

### 2.3. Slaughtering Procedure

At the end of the feeding trial, three male bucks were randomly selected from each of the three treatment groups of each breed and slaughtered at the Muscat Municipal Abattoir according to the Ministerial Resolution No. 255/2020 issuing the executive regulations of the Animal Welfare Law-Oman. During processing, individual carcass components including the head, skin, feet, fat (omental, mesenteric, and kidney), full, and empty alimentary tract, liver, spleen, heart, lung, and trachea were weighed and recorded. The weights of the full and empty reticulorumen were used to calculate the difference between full and empty body weights. The weight of the gut content was deducted from the slaughter weight to determine the empty body weight (EBW). The hot carcass weight (HCW) was recorded within an hour of the slaughter before the carcass was refrigerated to a temperature of 1–4 °C for 24 h.

### 2.4. Carcass Measurements

Following a 24 h chilling period, several external carcass measurements were collected. Leg length, gigot width, maximum shoulder width, depth from scapula to sternum, and width behind shoulders were all measured. For each carcass, all measurements were taken in cm using the methodology stated by Moxham and Brownlie [24]. Carcasses were separated vertically through the vertebrae using a control band saw. The carcass’s shoulder was cut at the caudal side of the seventh rib and then between the last and second-to-last lumbar vertebrae. A cut was made between the 12th and 13th ribs to separate the rack and the loin. The *m. longissimus dorsi* (LD) was taken from the loin region, its cross-sectional surface removed of fat, and it was then tested to determine its quality traits. Meanwhile, the *m. semitendinosus* (SM) was extracted from the leg and weighed to determine yield characteristics.

### 2.5. Meat Quality Evaluation

LD and SM were used to determine meat quality indicators such as ultimate ph, sarcomere length, expressed juice (drip loss), Warner–Bratzler shear force, cooking loss, and fresh color parameters (L*, a*, and b*), where L*, a*, and b* test relative lightness, relative redness, and relative yellowness, respectively [25]. The ultimate ph and chilled muscle samples were measured according to [22]. The lengths of sarcomeres were calculated by laser diffraction using a procedure described by [26]. The filter paper method was used to test the expressed juice as a total wetted area less than the meat area (cm^2^) relative to sample weight (g) as described by Kadim et al. [27]. For about 60 min, the fresh-cut surface of the meat was exposed to ambient air temperature to allow for light reflection for color measurements (L*, a*, and b*). The Minolta Chroma Meter CR-300 (Minolta Co., Ltd., Tokyo, Japan) was used to obtain these measurements, which has a color-measuring diameter of 1.1 cm.

### 2.6. Fatty Acid Analysis by Gas Chromatograph (GC)

The fatty acid composition of LD was analyzed and extracted using the AOAC [23] method for trans fats detection by gas chromatography with gas–liquid chromatography (GLC), a flame ionization detector (FID), and a helium (He) carrier due to using a capillary column. At a constant flow rate of 1.0 mL/min, ultra-high purity helium (He) with a purity of 99.9999% was used as a carrier gas. A 2 g subcutaneous fat sample was well mixed with 1 N KOH, and 5 mg of Tricosanoic acid (C23), an internal standard, was used. The fatty acid samples were analyzed and separated using the method described by Hernández [25]. The data were gathered using full-scan mass spectra with a scan range of 35–500 atomic mass units (amu). Each injected sample was 1 μL in volume, with a split ratio of 20:1. The oven temperature was set at 50 °C with no hold time, and then raised at a rate of 40 °C per minute up to 250 °C, where it was kept for 10 min. For identification of unknown substances, the spectra generated throughout this method were compared to existing mass spectrum libraries, notably NIST 2011 v.2.3 and Wiley’s 9th edition. Supelco 37 component FAME mixture (catalogue number 47885-U) was used for further verification. Total saturated fatty acids (Sfa), polyunsaturated fatty acids (Pufa), and desirable fatty acids (DFA) were calculated by adding the amounts of each fatty acid type, allowing for a complete assessment of their levels. In addition, the Pufa/Sfa ratio was calculated. Using tricosanoic acid (C23), as an internal reference, is a useful approach for quantifying fatty acids. It also enables an accurate measurement of fatty acids.

### 2.7. Statistical Analysis

Descriptive statistics were conducted for each trait. ANOVA was undertaken to assess SP influence on carcass characteristics, fatty acid profile, and meat quality traits of two Omani goat breeds among CON, T1, and T2 groups. The effects of SP, breed, and their interactions on the goats’ performance were determined using the general linear model procedure (GLM) of the statistical analysis system [28]. The data were subjected to factorial ANOVA analysis (PROC GLM), with breed, treatment levels, and their interactions being fitted as fixed effects, and goats’ performances as dependent variables (Equation (1)).
Y_ijk_ = μ + A_i_ + B_j_ + AB_ij_ + e_ijk_(1)
where Y_ijk_ is the goats’ performance of the lth kid in the ith breed, jth treatments, and ijth interaction; μ is the overall mean; A_i_ is the effect of the ith breed, i = 1 for Jabbali and 2 for Sahrawi; B_j_ is the effect of the jth treatment, j = 1 for CON, 2 for T1, and 3 for T2. Furthermore, ABij is the effect of ijth interaction between the ith breed and jth treatment, and eijk is the effect of the random error associated with the lth individual assumed to be normally distributed (0, Iσ^2^). All figures were drawn using Origin software (Origin, Version 2023. Origin Lab Corporation, Northampton, MA, USA) according to the previous statistical model. Differences between mean values of breeds and dietary treatments were obtained by Duncan’s multiple range test [29]. Differences were considered statistically significant at *p* < 0.05. The results are presented as least square means along with their standard errors (LSM ± SE).

## 3. Results

### 3.1. Animal Performance and Carcass Characteristics

No breed differences were detected in most growth performance and carcass characteristics traits, such as final body weight, carcass weight, dressing-out%, carcass dimensions, and carcass cut weights (Appendix A). An addition of SP in the Sahrawi bucks concentrate feed resulted in a better average daily gain (ADG) in both supplemented groups (85.5 ± 10.97) compared to the CON group (56.5 ± 5.38) (Figure 1a). There was clearly significant impact (*p* < 0.05) on the body length of Sahrawi bucks in T1 (43.30 ± 2.87) compared to the CON and T2 groups (39.10 ± 0.81 and 38.27 ± 0.52, respectively) (Figure 1b). Moreover, there was also a significant impact (*p* < 0.05) on the leg length of Sahrawi bucks in T1 (18.45 ± 1.49 cm) compared to T2 (16.60 ± 0.29 cm) (Figure 1c). In addition, a significant impact (*p* < 0.05) on the rack weight of Sahrawi goats was shown between T1 (1.54 ± 0.03) and CON (1.22 ± 0.04) (Figure 1d). All of these trends were not observed in Jabbali bucks. The only significant impact in Jabbali bucks was found on lion weight between T1 (1.06 ± 0.05) and T2 (0.98 ± 0.06) groups (Appendix A).

There were also no breed differences (*p* > 0.05) for all non-carcass components except for omental and kidney fat weights of both breeds and heart and kidney weights of Jabbali bucks (Appendix A). The weights of omental and kidney fat were significantly increased by SP compared to CON in both breeds (Figure 2a,b). The weights of the heart (0.36 ± 0.27 g) and kidney (0.29 ± 0.21 g) were significantly higher in CON compared to both SP groups (0.09 ± 0.01 and 0.06 ± 0.01) of the Jabbali breed, respectively.

### 3.2. Meat Quality

There were no effects of SP supplementation on all meat quality parameters in either longissimus dorsi (LD) or semitendinosus muscles (SM) (*p* > 0.05) except for ultimate ph and color L* (lightness) for Sahrawi bucks (Appendix A). For the LD, the T1 group (5.84 ± 0.17) had a significantly higher average ultimate ph value compared to the control (5.19 ± 0.09) and T2 (5.22 ± 0.09) groups in Sahrawi bucks (Figure 3a). The T2 group (46.55 ± 1.49) showed highly significant average meat color L* (lightness) value compared to the T1 group (41.52 ± 0.62) in the same breed (Figure 3b).

For the SM, the T1 group (6.30 ± 0.06) had significantly higher average ultimate ph value compared to the T2 group (5.73 ± 0.08) in Sahrawi bucks (Figure 4a). The T2 group (48.69 ± 0.65) showed highly significant average meat color L* (lightness) value compared to the T1 group (43.41 ± 0.82) in the same breed (Figure 4b).

### 3.3. Fatty Acid Profile

The fatty acid profiles of the LD muscle for Jabbali and Sahrawi bucks fed SP are presented in Appendix A. Some saturated fatty acids (C24:0), monounsaturated fatty acids (C16:1), and polyunsaturated fatty acid (C20:3n6) of the LD in Jabbali bucks showed significant differences from diets supplemented with SP. However, most saturated fatty acids (Sfa) except (C16:0 and C20:0), monounsaturated fatty acids (Mufa), polyunsaturated fatty acids (Pufa) except (C20:3n6), Pufa n-6, and Pufa n-3 and n-6/n-3 ratio of the LD in Sahrawi bucks showed significant differences to diets supplemented with SP (*p* < 0.05). In addition, the LD of Sahrawi goats fed diets supplemented with the SP of the T1 group (0.07 ± 0.01 and 0.49 ± 0.23) had significantly decreased (*p* < 0.05) amounts of pentadecanoic (C15:0) and margaric (C17:0) acids compared to the T2 (0.98 ± 0.26 and 2.52 ± 0.59) and CON (0.66 ± 0.19 and 1.91 ± 0.54, respectively) (Figure 5a,b).

## 4. Discussion

The current study revealed that supplemented SP in animal feed improved growth performance and some meat quality traits in two Omani goat breeds. This improvement could be attributed to an increase in carcass measurements, such as body length, leg length, and the rack weight, which is confirmed in our research. Average daily gain and carcass traits (body length, leg length, and rack weight) of Sahrawi bucks were significantly higher in the T1 group than in the CON group. However, the outcomes largely depend on several factors, such as the chemical composition, incorporation rate, microalgae species, and environmental conditions during growth [30].

The following interpretations can explain the increase in carcass measurements: (1) Sp can improve nutritional absorption through its function as a growth stimulus for gut pathogens [31]. (2) Adding SP in the diet may improve animal performance and meat quality [18,30], due to its high content of all essential amino acids and protein, up to 70% by dry weight [32]. (3) SP supplements provide goats with a premix of vitamins and minerals that are not often available in conventional diets. As reported by Moury et al. [33], vitamin supplementation is often not necessary when SP is added to animals’ diet. Adding a vitamins and minerals premix to an animals’ diet may affect their performance and carcass quality [34]. (4) In our previous research in the same breeds, goats fed SP had a lower feed conversion ratio (FCR). Adding SP to the concentration diet resulted in a considerably lower FCR value in the T1 group (7.70 ± 0.52) compared to the CON group (9.88 ± 2.49). In the Jabbali and Sahrawi goat breeds, the T2 group had a significant effect (*p* < 0.05), with FCR of (9.79 ± 0.61), whereas the CON group had a higher FCR value (15.15 ± 1.67) [17].

Moreover, Tovar et al. [35] stated that higher animal performance might be attributed to the high nutritional level of SP and its ability to result in the production of extracellular enzymes by the gut microbiota. In general, the increased growth performance in goats under a diet supplemented with SP might be attributed to the beneficial nutritional profile of SP. It is rich in minerals, vitamins, essential amino acids, fatty acids, and other nutrients that may increase the growth.

No differences between breeds were observed in terms of both hot and cold carcass weights. Similarly, Kalbe et al. [36] reported that there was no significant effect on hot and cold carcass weights when adding 7% and 5% of microalgae supplementation in a piglet diet and fattening diet, respectively. Additionally, the supplementation of SP in the diet in our study did not have a significant impact on dressing percentage for either breed. The Sahrawi breed showed the only significant difference regarding the effect of SP on body length in T1, possibly due to the non-diet related ability of this breed to climb mountains and its possessing of an appropriate phenotype.

The weights of the full and empty digestive tract were higher in both SP groups compared to the CON group, although these differences were not significant in both breeds. This study also showed an increase in fat deposition due to the incorporation of SP into the diet, with a significant increase in the weights of omental and kidney fat (*p* < 0.05) of both breeds. Mahgoub and Lu [37] also observed a similar trend in internal visceral fat in various breeds of goats.

Several factors impact meat quality, including breed, age, sex, weight, and nutrition [38]. The ultimate ph and color of the LD and MS are the most critical criteria in determining the quality of meat [39,40]. In the present study, the ultimate ph value of the T1 group was significantly higher than CON and T2 groups in LD than T2 group in the SM of Sahrawi bucks. The values of LD and SM color were non-significant for both breeds, except that the value of color lightness (L*) for group T2 was significantly higher than for group T1 in the LD and MS of Sahrawi bucks. The higher values of ultimate ph and color lightness (L*) of the LD and MS may be due to the antioxidant activity of the SP diet, which contained high carotenoids possibly deposited in the muscle [41,42]. The ultimate ph values observed here in goat carcasses fall within the normal range reported for SM 5.4–6.3 [43], but were found to be below the acceptable range for LD across breeds 5.6–5.8 [44]. The elevated ph readings in goat meat indicate that goats are more susceptible to stress in general [45,46]. The optimal ph measures can vary between species due to fiber type composition and physiology. The color of the meat is an essential quality characteristic since it is the first criterion that customers consider when selecting fresh meat [47], so having no impact from feeding SP in diets is beneficial.

Previous literature indicated that goat breeds and supplementation type can impact fatty acid profiles of goat meat [48]. In the current study, some saturated fatty acids (C24:0), monounsaturated fatty acids (C16:1), and polyunsaturated fatty acids (C20:3n6) of the LD in Jabbali bucks showed significant differences to diets supplemented with SP. However, all saturated fatty acids (Sfa) except (C16:0 and C20:0), monounsaturated fatty acids (Mufa), polyunsaturated fatty acids (Pufa) except (C20:3n6), Pufa n-6, and the Pufa n-3 and n-6/n-3 ratio of the LD in Sahrawi bucks showed significant differences compared to diets supplemented with SP (*p* < 0.05). The LD muscle of Sahrawi goats fed diets supplemented with SP of the T1 group significantly decreased in the amounts of pentadecanoic and margaric acids compared to the T2 and CON groups.

The current study determined that the levels of Pufa n-6 in the LD muscle of Jabbali goats given a microalgae diet were lower, but not significant (*p* = 0.307), due to their diet that included less amounts of C18:2Cisn6 than the CON group. This is in contrast with the findings in Sahrawi goats, in which both T1 and T2 were significantly higher than CON. Furthermore, the addition of Pufa n-3, particularly docosahexaenoic acid (DHA) present in microalgae, may inhibit the synthesis of longer-chain Pufa n-6 since they need the same elongate and desaturase enzymes [49]. On the other hand, the water-holding capacity (WHC) of pig muscle is correlated to certain fatty acids, particularly Pufa [50,51]. Furthermore, supplementing with Pufa n-3 tends to help muscle cells produce a flexible lipid bilayer membrane, which leads to an increase in their water-holding capacity (WHC).

Briolay et al. [52] and Jeromson et al. [53] observed an association between membrane restructuring, the subsequent effect on membrane protein function, and increased muscle protein synthesis in response to Pufa n-3 treatment. This conclusion is consistent with our finding that goats assigned a microalgae-based diet have higher protein content in the LD muscle. Furthermore, Wei et al. [54] showed that DHA supplementation promotes protein synthesis for growing pig muscle. SP has a potential dietary antioxidant supplement in the goat industry, which improves goats’ growth and muscular development. The dietary supplementation was helpful in producing meat with less fat and more Mufa [47].

In summary, the incorporation of SP into goat diets has a low effect on the fatty acid components of the muscle, in line with the previous reports on antioxidants for ruminant animals [55]. The increasing SP inclusion level is related to an increasing in the Mufa of the LD in Sahrawi bucks for C14:1, C16:1, C16:1 Cis9, and C17:1. While the C13:1 and C18:1 Cis (n9) content was not affected by SP diet for both breeds. This study ascertains the influence of dietary supplementation on muscle fatty acids. The impact of SP on Sahrawi goats was found to be more significant than on Jabbali goats. This difference may be attributed to the natural environment where the goats are raised. Jabbali goats are typically raised in the Al-Hajar Mountains, where the rumen microbiota community is not adapted to digest green algae. Therefore, the inclusion of SP in their diet may not have as significant an impact on their growth performance. In contrast, Sahrawi goats are commonly raised in the plains and valleys where the rumen microbiota community is adapted to digest green algae. This may explain why adding SP to their diet resulted in a more significant improvement in their product traits. Finally, the SP supplementation elicited the desired results within this study with a slight discrepancy between both breeds, possibly due to the variation in genetic composition for muscle growth across breeds.

## 5. Conclusions

The addition of microalgae SP at a dose of 2 g and 4 g/goat daily may enhance the daily gains, separate parts of carcass, meat quality, and fatty acid composition of Omani bucks reared under different environmental conditions, without any adverse effect on their physiological responses. However, this improvement was more significant for Sahrawi bucks compared with Jabbali bucks and depends on the level of nutritional supplementation, goat breed, and the natural environment in which the goats are raised. Finally, a study of rumen metagenomics using 16S ribosomal RNA (rRNA) is needed to compare the rumen microbiota of Jabbali and Sahrawi goats fed different levels of SP.

## Figures and Tables

**Figure 1 animals-13-02976-f001:**
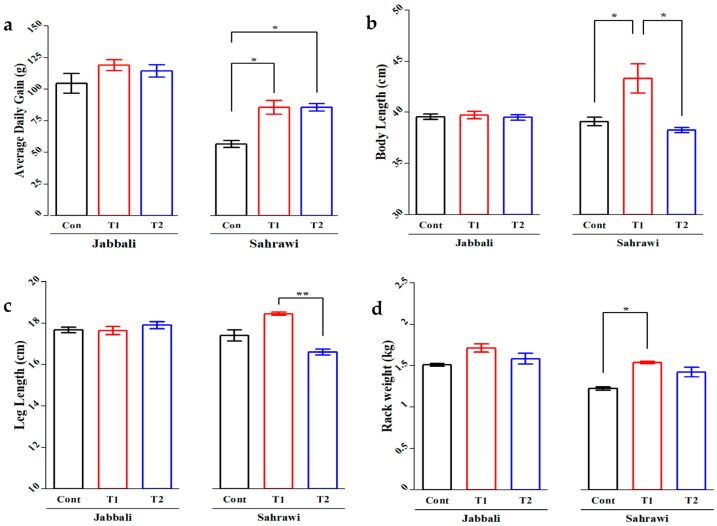
The effects of SP on (**a**) average daily gain (g), (**b**) body length (cm), (**c**) leg length (cm), and (**d**) rack weight (kg) of Jabbali and Sahrawi Omani goat breeds. The data are shown as least square means and standard errors. *p* < 0.05 is consider as significant. The * represents *p* < 0.05 and ** represent *p* < 0.01.

**Figure 2 animals-13-02976-f002:**
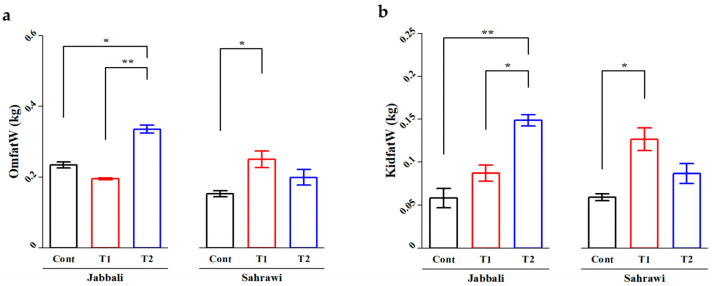
The effect of SP on weights of (**a**) omental and (**b**) kidney fat (g) of Omani Jabbali and Sahrawi goat breeds. The data are shown as least square means and their standard errors. *p* < 0.05 is consider as significant. The * represents *p* < 0.05 and ** represent *p* < 0.01.

**Figure 3 animals-13-02976-f003:**
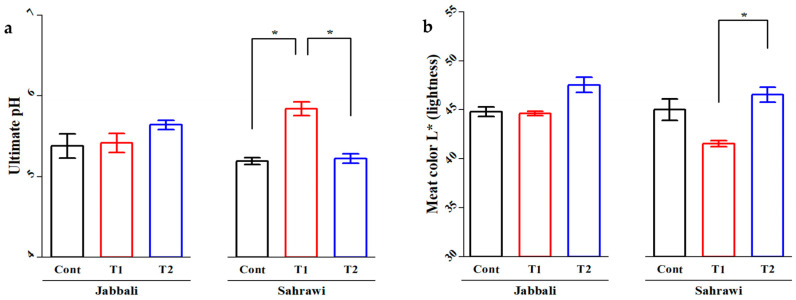
The SP effects on (**a**) ultimate ph and (**b**) meat color (L* lightness) of longissimus dorsi in Jabbali and Sahrawi breeds. The data are shown as least square means and standard errors. *p* < 0.05 is consider as significant. The * represents *p* < 0.05.

**Figure 4 animals-13-02976-f004:**
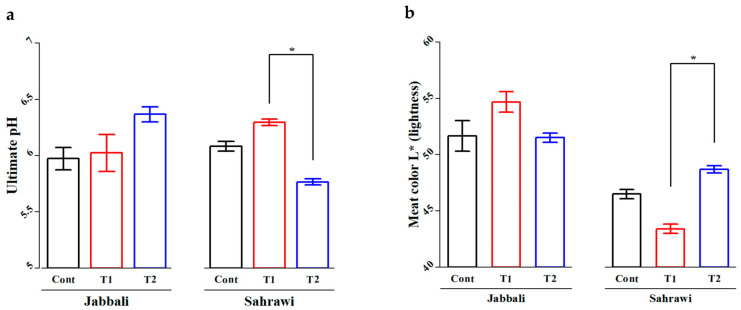
The SP effects on (**a**) ultimate ph and (**b**) meat color (L* lightness) of semitendinosus muscles in Jabbali and Sahrawi breeds. The data are shown as least square means and standard errors. *p* < 0.05 is consider as significant. The * represents *p* < 0.05.

**Figure 5 animals-13-02976-f005:**
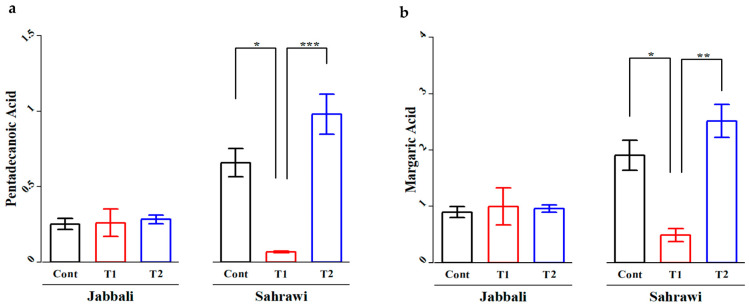
The effect of SP on (**a**) pentadecanoic and (**b**) margaric acids of LD of goat bucks. The * represent *p* < 0.05, ** represent *p* < 0.01, and *** represent *p* < 0.001.

**Table 1 animals-13-02976-t001:** Nutrient composition (g/100 g DM) and dry matter content (g/100 g fresh wt.) of SP and basal diet of Rhodes grass hay and concentrate.

Nutrients (In Dry Matter)	Concentrate	SP	Rhodes Grass Hay	Unit
Dry matter % (DM)	90.0	95.1	89.70	g/100 g Fresh Wt.
Crude protein (CP)	14.0	62.48	7.22	g/100 g DM
Crude fiber (CF)	9.8	2.9	34.3	g/100 g DM
Ether extract (EE)	2.5	1.05	1.00	g/100 g DM
Ash	9.2	7.55	9.80	g/100 g DM
Nitrogen free extract (NFE)	64.5	26.02	47.7	g/100 g DM
Neutral detergent fiber (NDF)	28.60	1.92	74.00	g/100 g DM
Acid detergent fiber (ADF)	11.42	0.37	46.7	g/100 g DM
Metabolisable energy (ME; MJ/kg DM) ^1^	11.97	11.63	8.30	kJ/100 g DM

Using the analyses expressed as g/kg DM gives ME (R) direct as MJ/kg DM. ^1^ ME (R) = 0.012 CP + 0.031 EE + 0.005 CF + 0.014 NFE.

## Data Availability

The data that support this study will be shared upon reasonable request to the corresponding author.

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
