# Peer review of "Effect of Spirulina platensis Supplementation on Carcass Characteristics, Fatty Acid Profile, and Meat Quality of Omani Goats"

_animals, 2023, doi:10.3390/ani13182976_

Round 1

Reviewer 1 Report (Previous Reviewer 1)

I have read Effect of Spirulina Platensis Supplementation on Carcass Characteristics, Fatty Acid Profile and Meat Quality of Jabbali and Sahrawi Omani Goats, and found some need improve.

The title need improve.

Abstract need improve, because it is a description of the results, not a summary. Other, L23 treatment 2?

1 Introduction is need improve, especially the aim of this paper.

2.5. Meat quality evaluation Why chose the LD and SM to determine meat quality indicators such as ultimate pH, sarcomere length, expressed juice (drip loss), Warner-Bratzler shear force, cooking loss, and fresh colour parameters?

2.7. Statistical analysis need improve.

L196-205 The sentences need improve.

L220 Figs 2. a and b?

L222 0.09±0.0 and 0.06±0.01?

....................................

L291-298 The sentences need improve.

L361-370 The sentences need improve.

Conclusion need improve.

Figures need improve, marked the a, b......

I have read Effect of Spirulina Platensis Supplementation on Carcass Characteristics, Fatty Acid Profile and Meat Quality of Jabbali and Sahrawi Omani Goats, and found some need improve.

The title need improve.

Abstract need improve, because it is a description of the results, not a summary. Other, L23 treatment 2?

1 Introduction is need improve, especially the aim of this paper.

2.5. Meat quality evaluation Why chose the LD and SM to determine meat quality indicators such as ultimate pH, sarcomere length, expressed juice (drip loss), Warner-Bratzler shear force, cooking loss, and fresh colour parameters?

2.7. Statistical analysis need improve.

L196-205 The sentences need improve.

L220 Figs 2. a and b?

L222 0.09±0.0 and 0.06±0.01?

....................................

L291-298 The sentences need improve.

L361-370 The sentences need improve.

Conclusion need improve.

Figures need improve, marked the a, b......

Author Response

Reviewer 2 Report (New Reviewer)

The article is clearly defined although there are some confusing paragraphs since it sometimes compares differences between breeds when only the effect of supplementation is being assessed.

Regarding to the material and methods a proximate composition of the supplementation must be included instead to the different composition of ingredients.

The different results only must be included once in table or in figure, but not in both forms. This is a repetition that should be avoided.  Also, there is a misunderstanding regarding to the C20:3n6, the authors considered this fatty acid as monounsaturated instead of PUFA one.

The findings should be discussed in more depth avoiding repeating the results obtained Many references are missing and some grammatical mistakes are found.

I should recommend a deeper revision and its publication as a short note.

Author Response

Reviewer 3 Report (New Reviewer)

This paper is focused on the evaluating the effect of Spirulina Plantesis supplementation on carcass characteristics, fatty acid content and meat quality of goats. The research idea is interesting and innovative considering the current trend of using microalgae in animal nutrition. However, the work has serious shortcomings regarding the interpretation of the obtained results which has to be severely improved. The impression is made that authors interpreted the results of other research ignoring their own results. Furthermore, some obtained results were misinterpreted.

P1, L42: Can you provide more recent data regarding goat meat production?

P3, Table 1: Provide chemical composition for all diets (control and two treatments).

P4, section 2.6: Provide the model of gas chromatography and detector you used in the study. Provide more information on type of gas carrier and flow you used? How did you express the results of fatty acids?

P5, Table 2: Provide full names for all abbreviated parameters below the table

P5, Table 2: Did you determine feed conversion ratio?

The same results of some parameters (leg length, body length, rack length) are presented in Table 2 and Fig.1. Please choose one way to present the obtained results. This also stands for Table 3/Fig.2; Table 4/Fig.3; Table 5/Fig.4; Table 6/Fig. 5.

P9, L262-263:  C20:3n6 is not monounsaturated fatty acid but polyunsaturated fatty acid.

P9, L265: C20:3n6 is not monounsaturated fatty acid.

P9, section 3.3: Authors should be focused on reporting the results on the most important FA and group of FA, particularly SFA, MUFA, n-6 PUFA, n-3 PUFA and n-6/n-3 ratio.

It would be useful to provide fatty acid profile of SP used in the study either in Table 1 or Table 6

Authors should indicate in Introduction section their previously published paper (doi: 10.1071/AN21483) on the use of spirulina on goat performance.

P10, L209-301: The sentence: “In the research by Al-Yahyaey et al. (2022), goats fed SP had a higher feed conversion ratio (FCR)”. This is not correct. In your previously published paper (Al-Yahyaey et al., 2022), goats fed SP had lower FCR than that goats fed CON.

P10, L300-302: The sentence: “adding SP to the concentrate diet resulted in a considerably higher FCR in the T1 group (7.70 ± 0.52) than in the CON group (9.88 ± 2.49).” FCR in T1 was lower than that in CON, not higher. Correct that.

P10, L302-304: The sentence: “the T2 group had a significant (P<0.05) effect, with an FCR of 9.79 ± 0.61 whereas the CON group had a less FCR of 15.15 ± 1.67.” CON had higher FCR than T2, not lower.

Generally, discussion regarding influence of spirulina on growth performance parameters should be severely modified. Authors were rather focused on citing other literature and interpretations of other assumptions than their own observations and conclusions supported by the obtained results.

P11, L321: Delete “The author additionally” from the sentence.

P11, L329: The reference Warner, 2015 is missing in the reference list.

P11, L348-349: The sentence “In this study, no significant differences (P>0.05) were detected in saturated fatty acids (SFA) among the treatments for the two breeds” is not quite correct. According to Table 6, total SFA for T2 was significantly higher than that in T1 for Sahrawi goats.

P11, L349-350: You stated “The LD muscle fat composition and lipid oxidation decreased in the supplemented groups compared to the control group (P<0.05). You did not determine the parameters of lipid oxidation to state such conclusion.

P11, L351-353: The sentence “Furthermore, despite the reported increase in subcutaneous and intermuscular adipose tissue deposition in the C130 versus the C260, internal fat depot accumulation remained similar”. The sentence is quite confusing. Please clarify it.

P11, L353-357: Authors discussed the changes in the amount pentadecanoic and margaric acid, instead of discussing changes in the level of some MUFA, essential n-6 and n-3 PUFA, n-6/n-3. Generally, the discussion related to changes in FA profile in Longissimus dorsi should be greatly enhanced.

P11, L355: “...the content of fatty acids decreased…” the content of which fatty acids? Specify it.

P11, L359: “…did not raise the rate of intramuscular fat in the Longissimus lumborum muscle”. Based on which results this conclusion was derived?

P11, L361: polyunsaturated fatty acids should be in lowercase

P11, L361-363: The sentence “The current study determined that the levels of Polyunsaturated Fatty Acids (PUFA) n-6 in the muscles of Jabbali goats given a microalgae diet were lower due to their diet included less 18:2 n-6 than the control group” is not correct. According to Table 6, the level of PUFA in Jabbali goats was not significantly (p=0.307) affected by dietary treatment. However, the level of PUFA in Sahrawi goats was significantly higher in both T1 and T2 than in CON.

P11, L367: is the reference tek et al., 2015  the same as  ÄŒitek et al., 2015?

P11, L366-370: it is unclear how the authors explain the relationship between water-holding capacity (WHC) and n-3 PUFA even though they did not determine WHC in their study.

P13, L378-379: Is the sentence “The dietary supplementation was helpful in producing meat with less fat and more MUFA” was the conclusion based on your results or some other research? If it is from your study, the statement is not correct as total MUFA was not significantly changed (Table 6). If it is cited from other study, please provide suitable reference.

P13, L379-380: You stated “Moreover, Spirulina added in the diet influence the lipid oxidation and stability in the content of MUFA increased”. I'm not entirely clear on how this fits into your results? You neither determine oxidative stability nor significant change in MUFA.

P12: Conclusion section should be improved and in line with those in Abstract section.

P14, L530: The reference Wan 2016 should be cited according to the journal instruction.

There are some minor typing errors observed.

Round 2

Reviewer 1 Report (Previous Reviewer 1)

Effect of Spirulina Platensis Supplementation on Carcase Characteristics, Fatty Acid Profile and Meat Quality of Omani Goats.

I have read the paper, the authors have good reviewed, I suggested accept.

Some sentences need improve.

Author Response

Reviewer 2 Report (New Reviewer)

After the revision I consider that the results only must be included once in table or in figure, but not in both forms so the the figures shoul be removed. 

Author Response

This manuscript is a resubmission of an earlier submission. The following is a list of the peer review reports and author responses from that submission.

Round 1

Reviewer 1 Report

This was studied to the Effect of Spirulina Platensis Supplementation on Carcase Characteristics, Fatty Acid Profile and Meat Quality of Omani Goats, I have read the paper carefully,English need improve, and it has some question to answer.

Title: need review. In this paper, the goat of Jabbali and Sahrawi were used.

Jabbali and Sahrawi are both the content of con, T1 and T2, suggested improve.

Abstract: need support with data, please improve.

L37-63 The sentences need rewrite. Some information are old. Such as Red meat is commonly consumed in many parts of the world with goat meat consumption rising globally by 41.66% between 2000-2012, I think 2020-2022 is good.

L44 diets  (Adeyemi et al., 2016)?

L72-77 The aim of this paper is not clear.

How to measure the Ultimate pH? Especially, time?

L135 fresh colour L*, a* and b* or fresh colour (L*, a* and b*)?

cm2 or cm2?

Statistics Analysis: please rewrite.

Discussion need improve.

L300-306 The sentences need rewrite.

Table 4. need improve.

Figure 4. Where is Ultimate pH?

All the figures need improve.

Conclusions need rewrite.

Reviewer 2 Report

The manuscript entitled “Effect of Spirulina Platensis Supplementation on Carcass Characteristics, Fatty Acid Profile and Meat Quality of Omani Goats” summarized the establish feeding strategies that would enhance goat meat production by incorporating SP into goat diets and evaluating its effect on fatty acid profile and meat quality..  Generally, the manuscript is poor. This paper has several weaknesses and needs improvement

This manuscript has major language problems. There are too many for me to modify them all. Authors are strongly encouraged to seek a native English speaker who may assist you modifying the document.

Comments:

1.                The animals were slaughtered through exsanguination; they have not been stunned before?

2.                Were animals "reared" in an environmentally controlled unit? The manuscript makes claim about uniform temperature and humidity. This needs to be clarified and justified. How might this influence measurement of production traits?

3.                Why did the authors choose this technology instead of other GWAS? Please provide the reason.

4.                The abstract is not particularly informative and would benefit from more background.

5.                Summarize the abstract, focus on the main findings and mention the small conclusion in at the end of abstract

6.                In the Introduction focus on the objectives and insert a few new reference and relevant findings

7.                In material and method sections, references are missing.

8.                Most of the references mentioned are old and I suggest adding recent references, and the manuscript should be edited accordingly.

9.                I suggest the cite following paper in introduction part For more information you can read below reference

RNA-Seq reveals the potential molecular mechanisms of bovine KLF6 gene in the regulation of adipogenesis. International Journal of Biological Macromolecules, 195, 198-206.

10.              Material and method needs to clarifying and summarizing- some detailed needs

11.              The subtitles in the material and method needs to summarizing Ethical approval and references must be mentioned in M&M

12.              In the result section, the results were written very poorly. It should be written again and try to avoid a brief introduction in the starting of every result.

Author Response

Please see the attached document—also, the paper was assessed by a native English speaker Dr. Russell and Dr. Melanie from Australia who modified and edited the language. 

Thanks for the valuable feedback but it seems some comments not related to my work.

Reviewer 3 Report

Introduction

The introduction needs to be rewritten in order to make the objective and hypothesis of the work clearer

Material and methods

Line 91 = The factorial design is not an experimental design, it represents the way treatments are distributed in the experimental unit.

   - Describe the experimental design of the study.

  - Describe the factorial arrangement. If there are 2 breeds and 3 treatments, your factorial arrangement should be 2x3

Results

I recommend removing the CV column from the tables, so the table is cleaner.

Line 183- Did the ADG show a significant effect or a trend? rewrite the sentence

NOTE: Do not present the results as a trend, since the trend p values were not presented to the readers

The results must be rewritten, to make it easier to understand.

       - There is no need to keep informing that you used the GLM procedure in the results (e.g., Line 205)

       - Present the results, only with p-value, you don't need to present the average values they are already in the table

Discussion

It must be rewritten, the focus must be on explaining the results of the work and not comparing it with the results in the literature.

conclusion

It must be rewritten. The conclusion should present the main results of the study. The way it is now is just a continuation of the discussion.

Round 2

Reviewer 1 Report

I have read the paper carefully, and found the language also need improve.

An important question was not answer: Some information are old. Add the references in 2020-2023. only one 2022, two 2020, other references is 2019..............1973. Otherwise, I will reject.

Reviewer 2 Report

Reject (article has serious flaws, additional experiments needed, research not conducted correctly)

Reviewer 3 Report

In the general context, the discussion should be improved and in order to avoid excessive comparison of its results with the literature, but rather use the literature to explain the mechanisms involved in the results observed in their work.

Tables are still very difficult to assess

Introduction

Line 45-47: Paragraph is confusing, canola and palm oil have no antioxidant activity - please rewrite paragraph

M&M's

Line 100-101: Present the used line clearly

NOTE: As different levels are being evaluated (0, 2 and 4), the correct thing is that in addition to the average test, the orthogonal contrast analysis is carried out

Results

All tables are loaded with information, which makes their evaluation difficult. I am submitting a table model suggestion.

Table 1: there was an effect for loin weight for the Jabbali breed and the mean test was not performed